# Radiological findings in nontuberculous mycobacterial pulmonary diseases: A comparison between the *Mycobacterium avium* complex and the *Mycobacterium abscessus* complex

Hiroaki Nagano[1☺]*, Takeshi Kinjo[2☺], Jiro Fujita[2], Tomoo Kishaba[1]

1 Department of Respiratory Medicine, Okinawa Chubu Hospital, Okinawa, Japan, 2 Department of Infectious, Respiratory, and Digestive Medicine, Graduate School of Medicine, University of the Ryukyus, Okinawa, Japan

☺ These authors contributed equally to this work.
* hiroakinoko322violin@gmail.com

**Data Availability Statement:** All relevant data are within the paper.

## Abstract

The *Mycobacterium abscessus* complex (MABC) comprises rapidly growing mycobacteria and has received increasing attention recently, with an increasing number of associated infections reported worldwide. However, the clinical features of MABC pulmonary disease (MABC-PD), especially in terms of the chest computed tomography (CT) findings, are not fully understood. Thus, this retrospective, cross-sectional study aimed to evaluate the clinical background and chest high-resolution CT (HRCT) findings of MABC-PD in comparison with those of *Mycobacterium avium* complex PD (MAC-PD). Accordingly, 36 patients with MABC-PD and 65 patients with MAC-PD (defined according to the American Thoracic Society criteria), who were newly diagnosed at four major hospitals in Okinawa (Japan) between January 2012 and December 2017, were analyzed. With respect to their clinical background, only cardiovascular diseases were significantly more common in patients with MABC-PD than in those with MAC-PD (38.9% vs. 18.5%, p = 0.0245). HRCT revealed a significantly higher incidence of low attenuation in patients with MABC-PD than in those with MAC-PD (63.9% vs. 10.8%, p<0.0001). On analyzing only never-smokers (20 and 47 patients with MABC-PD and MAC-PD, respectively), this significant difference remained (65.0% vs. 8.5%, p<0.0001), suggesting MABC infection itself caused low attenuation. In terms of the distribution of abnormal shadows, the involvement of the right lower, left upper, and left lower lobes was more common in patients with MABC-PD than in those with MAC-PD. Furthermore, the mean number of involved lung lobes was significantly higher in patients with MABC-PD than in those with MAC-PD (5.6 vs. 4.7, p<0.001). Although further studies are needed, we assume that the aforementioned radiological features of MABC-PD are due to the high virulence of MABC.

**Funding:** The authors received no specific funding for this work.

**Competing interests:** The authors have declared that no competing interests exist.

## Introduction

Nontuberculous mycobacteria (NTM) are acid-fast bacteria; they are ubiquitous and can cause a variety of infections in humans. NTM pulmonary disease (NTM-PD) is the most common form of NTM infection. Its incidence continues to increase worldwide; this is true even in Japan due to an increasingly aging population and an increased awareness of the disease [1–3]. NTM comprise approximately 200 species, and the treatment strategy differs by the species; therefore, the first step to care for patients with NTM-PD is the identification of the causative NTM species from respiratory specimens. NTM-PD is most commonly caused by *Mycobacterium avium* complex (MAC). Rapidly growing mycobacteria such as *M. fortuitum*, *M. chelonae*, and those from the *M. abscessus* complex (MABC) are uncommon pathogens of NTM-PD; however, MABC has been frequently identified as the causative pathogen in patients with NTM-PD in South Korea and Taiwan [3–6]. In 2017, our data suggested that Okinawa (located in the southernmost region of Japan) was also one of the rare regions where MABC was the predominant cause of NTM-PD [7]. Studies on the clinical features of MABC-PD are insufficient, and only a few small-scale studies have assessed the characteristic computed tomography (CT) findings of MABC-PD [8, 9]. In clinical settings, the identification of the causative NTM species from respiratory specimens often takes weeks or months; thus, understanding the patient's background and radiological features that are suggestive of the causative NTM species is important for managing NTM-PD. Accordingly, the aim of this study was to evaluate the clinical features, especially the high-resolution CT (HRCT) findings, specific to MABC-PD. This was accomplished by comparing patients with MABC-PD and those with MAC-PD; because MAC is the most common cause of NTM-PD worldwide, it is often used as a comparative control in NTM studies. The manuscript details the methodology undertaken to achieve the study aim, key findings obtained, and relevant discussion with respect to previous literature.

## Materials and methods

### Patients

In this study, we included patients who were newly diagnosed with MABC-PD (the MABC-PD group) or MAC-PD (the MAC-PD group) between January 2012 and December 2017 at the Okinawa Chubu Hospital (550 hospital beds), University of the Ryukyus Hospital (600 hospital beds), Okinawa Hokubu Hospital (327 hospital beds), and Naha City Hospital (470 hospital beds). The diagnosis was made in accordance with the American Thoracic Society criteria [10]. The patients' medical records were retrospectively reviewed to compare the clinical backgrounds and chest HRCT findings between the two groups.

The exclusion criteria were as follows: 1) patients with a history of infection with other NTM species, 2) patients co-infected with respiratory pathogens other than NTM at the time of HRCT, 3) patients receiving antimycobacterial treatment at the time of HRCT, 4) patients with severe lung destruction secondary to underlying lung diseases and in whom an appropriate evaluation of NTM-PD-associated lung abnormalities was deemed impossible, and 5) patients with a history of lobectomy, total pneumonectomy, and tracheostomy or those who underwent mechanical ventilation procedures.

For NTM identification, respiratory specimens were cultured on 2% Ogawa agar, and the bacterial colonies obtained were collected for species identification. Species identification was performed via a DNA-DNA hybridization method using a commercially available identification kit (Kyokuto Pharmaceutical Industrial Co., Ltd., Tokyo, Japan).

## Chest HRCT evaluation

Chest HRCT scans taken on the day closest to the date of NTM diagnosis (1 year before and after diagnosis) were evaluated by two experienced chest physicians (Nagano and Kinjo). The radiological patterns of NTM-PD were classified into the following four types: nodular bronchiectatic (NB) type with cavity, NB type without cavity, fibro-cavitary (FC) type, and unclassified. The NB type was characterized by bilateral bronchiectasis with nodular infiltrates involving the middle lung zones, while the fibro-cavitary type was characterized by cavitary lesions typically located in the upper lobes [11, 12]. For each patient, the presence of patterns reflective of parenchymal abnormalities in the following six regions of the lung was recorded: right upper lobe, right middle lobe (RML), right lower lobe (RLL), left upper segment, left lingular segment (LLS), and left lower lobe (LLL). These patterns were based on previous reports [8, 9, 13, 14], and comprised the following: (1) centrilobular ground-glass opacity (GGO), (2) centrilobular nodules, (3) small nodules <10 mm, (4) nodules sized 10–29 mm, (5) tree-in-bud appearance, (6) volume loss, (7) cavity, (8) consolidation, (9) bronchiectasis, (10) low attenuation, (11) GGO, (12) linear scarring, and (13) calcification. Patients could present with more than one pattern.

## Statistical analysis

Nominal and continuous variables were compared between the MABC-PD and MAC-PD groups using the Fisher's exact test and the Wilcoxon/Kruskal–Wallis test, respectively. A p value < 0.05 was considered significant. All data were analyzed with JMP pro 15 (SAS Institute Inc., North Carolina, USA).

## Ethics

The Institutional Ethics Committee of the Okinawa Chubu Hospital approved this study (approval number: 2018–89). The need for informed consent from each patient for inclusion in this study was waived due to the study's retrospective nature. Even then, patients were given the opportunity to opt-out via the Okinawa Chubu Hospital's website.

## Results

The MABC-PD and MAC-PD groups comprised 36 and 65 patients, respectively. MAC consisted of *M. avium* (n = 16, 24.6%) and *M. intracellulare* (n = 49, 75.4%). On comparing the background characteristics between the two groups, it was found that the incidence of cardiovascular disease was higher in the MABC-PD group than in the MAC-PD group (38.9% vs. 18.5%, p = 0.025; Table 1).

In terms of the HRCT findings, there were no significant differences between two groups regarding the NB type with cavity and FC type. However, the NB type without cavity was more common in the MAC-PD group than in the MABC-PD group (66.2% vs. 38.7%, p = 0.008). Conversely, the unclassified type was more common in the MABC-PD group than in the MAC-PD group (30.6% vs. 12.6%, p = 0.0246; Table 2).

Furthermore, low attenuation was observed more commonly in the MABC-PD group than in the MAC-PD group (63.9% vs. 10.8%, p<0.0001). To exclude the effect of smoking, we analyzed the incidence of low attenuation among never-smokers (MABC-PD: 20 patients, MAC-PD: 47 patients). Accordingly, low attenuation was still found to be more common in the MABC-PD group than in the MAC-PD group (65.0% vs. 8.5%, p<0.0001). In terms of the distribution of abnormal shadows, the involvement of the RLL, left upper segment, and LLL was more common in the MABC-PD group than in the MAC-PD group. Furthermore, the

**Table 1. Patients' background characteristics.**

| | MABC-PD (n = 36) | MAC-PD (n = 65) | p value |
|---|---|---|---|
| **Age (median, years)** | 77 | 78 | 0.2487 |
| **Male sex** | 17 (47.2%) | 20 (30.8%) | 0.1317 |
| **BMI (median, kg/m²)** | 19.7 | 19.2 | 0.3815 |
| **Smoking history** | 14 (41.2%) | 17 (26.6%) | 0.1727 |
| **Comorbidities** | | | |
| **Interstitial lung disease** | 5 (13.9%) | 5 (7.7%) | 0.3225 |
| **Old healed tuberculosis** | 7 (19.4%) | 8 (12.3%) | 0.3867 |
| **COPD** | 7 (19.4%) | 7 (10.8%) | 0.2433 |
| **Bronchial asthma** | 8 (22.2%) | 9 (13.9%) | 0.4052 |
| **Gastroesophageal disease** | 4 (11.1%) | 3 (4.6%) | 0.2435 |
| **Lung cancer** | 1 (2.8%) | 1 (1.5%) | 1.0000 |
| **Other solid cancers** | 5 (13.9%) | 6 (9.2%) | 0.5152 |
| **Hematological cancer** | 2 (5.6%) | 1 (1.5%) | 0.2886 |
| **Cardiovascular diseases** | 14 (38.9%) | 12 (18.5%) | 0.0329 |
| **Chronic liver disease** | 5 (13.9%) | 2 (3.1%) | 0.0939 |
| **Chronic kidney disease** | 7 (19.4%) | 11 (16.9%) | 0.7898 |
| **Cerebrovascular disease** | 4 (11.1%) | 13 (20.0%) | 0.2842 |
| **Neuromuscular disease** | 1 (2.8%) | 1 (1.5%) | 1.0000 |
| **Autoimmune disease** | 4 (11.1%) | 9 (13.8%) | 0.7668 |
| **Diabetes mellitus** | 9 (25.0%) | 7 (10.8%) | 0.0869 |
| **Corticosteroid usage[#]** | 7 (19.4%) | 10 (15.4%) | 0.5921 |
| **Immunosuppressant usage** | 2 (5.6%) | 2 (3.1%) | 0.6149 |

[#] Patients receiving corticosteroid daily at any dose.

Abbreviations: MABC-PD, *Mycobacterium abscessu*s complex-pulmonary disease; MAC-PD, *Mycobacterium avium* complex-pulmonary disease; COPD, chronic obstructive pulmonary disease; BMI; body mass index.

mean number of involved lung lobes was significantly higher in the MABC-PD group than in the MAC-PD group (5.6 vs. 4.7, p <0.001; Table 3).

## Discussion

This study aimed to investigate the clinical features of MABC-PD by comparing them with those of MAC-PD. Although patients with NTM-PD co-infected with multiple NTM species have been described previously [15–17], the present study only included patients with NTM-PD who were infected with either MABC or MAC alone. Therefore, this study could compare the clinical features purely between MABC-PD and MAC-PD. Our data

**Table 2. General classification of the imaging findings.**

| | MABC-PD (n = 36) | MAC-PD (n = 65) | p value |
|---|---|---|---|
| **NB type with cavity** | 7 (19.4%) | 9 (13.9%) | 0.5710 |
| **NB type without cavity** | 14 (38.7%) | 43 (66.2%) | 0.0117 |
| **FC type** | 3 (8.3%) | 5 (7.7%) | 1.0000 |
| **Unclassified** | 11 (30.6%) | 8 (12.6%) | 0.0337 |

Abbreviations: MABC-PD, *Mycobacterium abscessu*s complex-pulmonary disease; MAC-PD, *Mycobacterium avium* complex-pulmonary disease; NB, nodular bronchiectatic; FC; fibro-cavitary.

**Table 3. Detailed comparison of the imaging findings between the MABC-PD and MAC-PD groups.**

| | MABC-PD (n = 36) | MAC-PD (n = 65) | p value |
|---|---|---|---|
| **Findings** | | | |
| **Centrilobular GGO** | 16 (44.4%) | 28 (43.1) | 1.0000 |
| **Centrilobular nodule** | 18 (50.0%) | 33 (50.5%) | 1.0000 |
| **Small nodule <10 mm** | 28 (77.8%) | 58 (89.2%) | 0.1484 |
| **Nodule of size 10–29 mm** | 10 (27.8%) | 14 (21.5%) | 0.4770 |
| **Tree-in-bud appearance** | 31 (86.1%) | 47 (72.3%) | 0.1409 |
| **Decreased volume** | 11 (30.6%) | 32 (49.2%) | 0.0930 |
| **Smooth-wall cavity** | 2 (5.6%) | 5 (7.7%) | 1.0000 |
| **Irregular-wall cavity** | 9 (25.0) | 8 (16.8%) | 0.1632 |
| **Consolidation** | 23 (63.9%) | 45 (69.2%) | 0.6596 |
| **Bronchiectasis** | 31 (86.1%) | 59 (90.8%) | 0.5152 |
| **Low attenuation** | 23 (63.9%) | 7 (10.8%) | < 0.0001 |
| **GGO** | 20 (55.6%) | 27 (41.5%) | 0.2136 |
| **Linear opacity** | 32 (88.9%) | 61 (93.9%) | 0.4507 |
| **Calcification** | 12 (33.3%) | 28 (43.1%) | 0.3987 |
| **Involved lung area** | | | |
| **RUL** | 33 (91.7%) | 55 (84.6%) | 0.3699 |
| **RML** | 31 (86.1%) | 54 (83.1%) | 0.7820 |
| **RLL** | 36 (100%) | 49 (75.4%) | 0.0005 |
| **LUS** | 35 (97.2%) | 47 (72.3%) | 0.0014 |
| **LLS** | 33 (91.7%) | 54 (83.1%) | 0.3679 |
| **LLL** | 35 (97.2%) | 50 (76.9%) | 0.0088 |
| **Mean number of involved lung lobes/segments** | 5.6 | 4.7 | 0.0005 |

Abbreviations: MABC, *Mycobacterium abscessus* complex; MAC, *Mycobacterium avium* complex; GGO, ground glass opacity; RUL, right upper lobe; RML, right middle lobe; RLL, right lower lobe; LUS, left upper segment; LLS, left lingular segment; LLL, left lower lobe.

demonstrated that cardiovascular diseases were more common in patients with MABC-PD than in those with MAC-PD. HRCT analysis revealed that low attenuation was significantly more common in MABC-PD than in MAC-PD; this significant difference was also noted among never-smokers. Additionally, the number of involved lung lobes was significantly higher in patients with MABC-PD than in those with MAC-PD.

Only few reports are available on an association between MABC and heart disease. Hsu et al. pointed out that patients with either cardiovascular diseases or risk factors for cardiovascular diseases (male sex and age ≥55 years) were 4.5 times more likely to have MABC-PD [18]. Moreover, the chronic inflammation associated with NTM-PD may induce a secondary cardiovascular disease [19]. Because the association between MABC and cardiovascular disease is not well understood, further studies are needed.

We first reported that low attenuation in the lungs is commonly seen in MABC-PD. This finding was persistent among never-smokers, indicating that MABC (and not smoking) might cause the low attenuation. No previously published reports have mentioned the relationship between MABC-PD and low attenuation. Kubo et al. analyzed CT findings obtained during the inspiration and expiration phases, and reported that air trapping occurred in the apical airways in MAC-PD [20, 21]. The authors indicated that lung function tests revealed air trapping-associated obstructive disorders in patients with MAC-PD. Although we did not perform CT during the inspiration and expiration phases, the low attenuation was considered to be an

important finding as it corresponded to air trapping on expiration [22]. Fujita et al. demonstrated that pathological findings (bronchiectasis and centrilobular nodules) in patients with MAC-PD indicated widespread lymphocyte infiltration and continuous epithelial cell infiltration from the bronchioles to the acinus, as well as the narrowing of the lumens of the bronchioles at various levels [23, 24]. These findings explained the obstructive respiratory dysfunction in patients with MAC-PD, which has been previously reported by Kubo et al. [20, 21]. There are few reports on the relationship between radiological findings and the pathology of MABC. However, by the same mechanisms as in MAC-PD, it is possible that intense inflammation in the airspace leads to a narrowing of the bronchiole lumen and air trapping in MABC-PD, which are then reflected as low attenuation on HRCT.

Findings from a typical case of low attenuation in MABC-PD are illustrated in Fig 1. The patient was a 63-year-old woman without prior lung infection or lung diseases. She had never smoked tobacco. Nevertheless, chest radiography revealed apparent diaphragm flattening and hyperinflation bilaterally (Fig 1A). Furthermore, chest HRCT revealed low-density areas in the lung parenchyma, despite no history of smoking or emphysema. In addition, this patient demonstrated multi-lobe involvement, including the right middle lobe, lingula, and bilateral lower lobes (arrows in Fig 1B and 1C).

Furthermore, the number of involved lung lobes was significantly higher in patients with MABC-PD than in those with MAC-PD. Although a few studies have compared the CT findings between MABC-PD and MAC-PD, there is no evidence that MABC-PD tends to involve more lung areas [8, 16]. Some studies reported that radiological imaging in MABC-PD revealed a more extensive distribution of abnormal patterns, including cavities, bronchial dilatation, and infiltration shadows [25, 26]. Fujita et al. reported that the RML and LLS are commonly involved in respiratory infections by MAC, and that centrilobular nodules and diffuse bronchiectasis are the characteristic radiological findings [23]. Our study demonstrated that

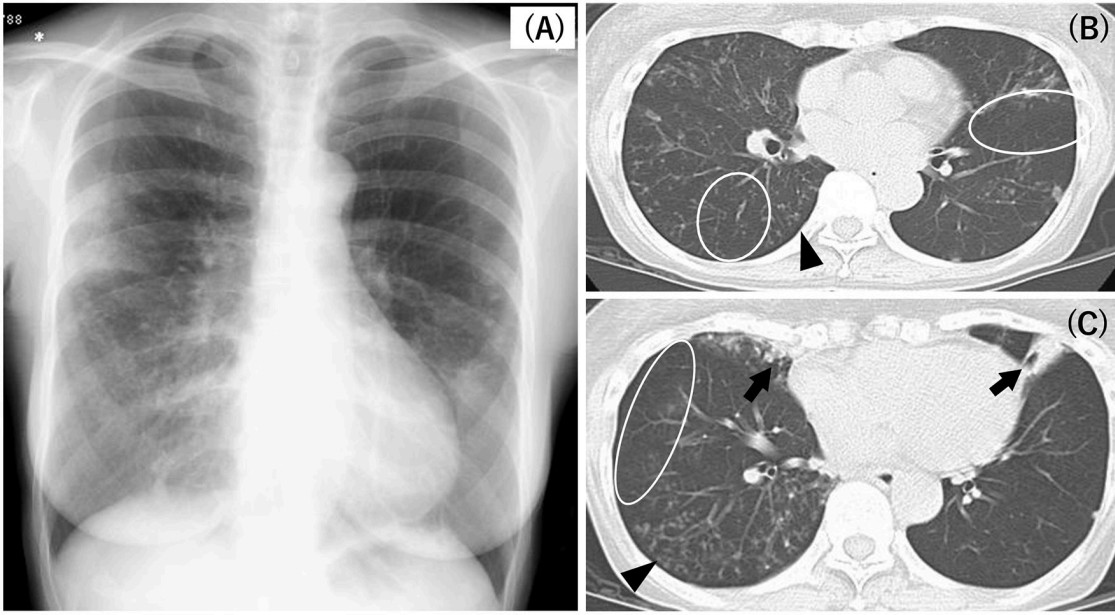

**Fig 1. A typical case of low attenuation in MABC-PD.** Chest radiograph (A) demonstrates bilateral diaphragm flattening and hyperinflation. Chest HRCT scans (B, C) reveal low-density areas in the lung parenchyma (oval enclosure mark); multi-lobe involvement with consolidation, bronchiectasis, and GGO (arrow) in the right middle lobe and LLS; and a tree-in-bud appearance and small centrilobular nodules in the RLL (arrowhead).

not only the RML and LLS, but also the right upper lobe, RLL, and LLL were involved in patients with MABC-PD. According to Chung et al., no significant difference was observed in the presence of small nodules, the tree-in-bud pattern, and bronchiectasis between MABC-PD and MAC-PD [8]. However, nodules, airspace consolidation, and thin-wall cavities were observed more commonly in patients with MAC-PD. Harada et al. reported that the nodular bronchiectatic form was commonly associated with *M. abscessus* than with *M. massiliense* [27]. Victoria et al. reported that MABC could be rapidly progressive and lead to lung destruction in a short period of time [28]. Although further investigation is needed, it is possible that the intensity of MABC virulence is related to the extensive distribution of lung lesions.

Our study has some limitations. First, as a retrospective study conducted in several hospitals, our data are sparse and potentially biased. Second, we could not analyze MABC at the subspecies level (*M. abscessus* subsp. *abscessus*, *M. abscessus* subsp. *massiliense*, and *M. abscessus* subsp. *bolletii*) because we could not preserve the bacterial colonies cultured from patients with MABC-PD. Differentiation between the three subspecies of MABC may reveal more profound results.

## Conclusions

Our data revealed that low attenuation and extensive lesions in the lungs on HRCT, possibly reflecting the high virulence of MABC, were more common in patients with MABC-PD than in those with MAC-PD. Understanding the radiological features of MABC-PD is important from a clinical perspective; thus, further radiological studies are needed to deepen our knowledge about MABC-PD.

## Acknowledgments

The authors thank Mariko Teruya, Akiko Maeda, and Kazuhiko Matsuno for their contribution to data collection. The authors also thank Dr. Joel Branch and Editage (www.editage.com) for English language editing.

## Author Contributions

**Conceptualization:** Hiroaki Nagano, Takeshi Kinjo, Jiro Fujita, Tomoo Kishaba.

**Data curation:** Hiroaki Nagano, Takeshi Kinjo.

**Formal analysis:** Takeshi Kinjo.

**Investigation:** Hiroaki Nagano.

**Methodology:** Hiroaki Nagano.

**Project administration:** Hiroaki Nagano.

**Software:** Takeshi Kinjo.

**Supervision:** Takeshi Kinjo, Jiro Fujita, Tomoo Kishaba.

**Writing – original draft:** Hiroaki Nagano.

**Writing – review & editing:** Hiroaki Nagano, Takeshi Kinjo, Jiro Fujita, Tomoo Kishaba.

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
