## [Decision Letter · Decision Letter 0]

18 Mar 2022

PONE-D-21-40311The radiological findings of nontuberculous mycobacterial pulmonary diseases: comparison between Mycobacterium avium complex and Mycobacterium abscessus complex.PLOS ONE

Dear Dr. Nagano,

Thank you for submitting your manuscript to PLOS ONE. After careful consideration, we feel that it has merit but does not fully meet PLOS ONE’s publication criteria as it currently stands. Therefore, we invite you to submit a revised version of the manuscript that addresses the points raised during the review process.

We look forward to receiving your revised manuscript.

Kind regards,

Thomas Byrd

Academic Editor

PLOS ONE

Journal Requirements:

(The funders had no role in study design, data collection and analysis, decision to publish, or preparation of the manuscript.)

4. Thank you for submitting the above manuscript to PLOS ONE. During our internal evaluation of the manuscript, we found significant text overlap between your submission and the following previously published works, some of which you are an author.

- https://journals.plos.org/plosone/article?id=10.1371%2Fjournal.pone.0186826

Please revise the manuscript to rephrase the duplicated text, cite your sources, and provide details as to how the current manuscript advances on previous work. Please note that further consideration is dependent on the submission of a manuscript that addresses these concerns about the overlap in text with published work.

Additional Editor Comments:

The investigator’s report differences in radiographic findings in hospitalized patients found to have sputum cultures positive for either MABC or MAC.

The investigators need to address the concerns of the reviewer and provide additional information regarding the patients in the study as well as context for their findings that relate to biologic plausibility. The following need to be addressed:

1) Do the increased areas of low attenuation found in patients with MABC lung infection suggest that patients with COPD/emphysema may be more susceptible to MABC lung infection compared to MAC? Please comment in the manuscript. The results of pulmonary function tests would be helpful in this regard – it would seem that this information should be available from a retrospective chart review from at least some of the patients and needs to be provided. If COPD/emphysema is more common, what would be the basis for this?

2) Why do the investigators postulate that more lung lobes are involved in patients found to have MABC lung infection? Furthermore, why do they postulate that cardiovascular disease was more common in patients with MABC cultured from their sputum?

3) What are differences comparing MABC to MAC in terms of known characteristics/virulence determinants that could account for reported radiographic observations?

4) How many patients underwent antibiotic treatment as a result of the concern for pulmonary NTM infection? This would be an indication of the treating physician’s concern that the patient had active, progressive NTM lung infection. Although IDSA and ATS provide guidelines, the judgment of the clinician seeing the patient has relevance to the significance of the radiographic findings reported in the study. In addition, at least some follow up information with time points after initiation of treatment should be provided on all patients. It is emphasized that the treatment status of these patients is important to judge the significance of the radiographic findings.

Reviewers' comments:

Reviewer's Responses to Questions

**Comments to the Author**

1. Is the manuscript technically sound, and do the data support the conclusions?

Reviewer #1: Partly

2. Has the statistical analysis been performed appropriately and rigorously? 

Reviewer #1: N/A

3. Have the authors made all data underlying the findings in their manuscript fully available?

Reviewer #1: Yes

4. Is the manuscript presented in an intelligible fashion and written in standard English?

Reviewer #1: Yes

5. Review Comments to the Author

Reviewer #1: Dear the authors,

The authors present the interesting findings about the differences between MAC and MABC pulmonary disease, focusing on CT findings. Although it is important to know the characteristics of MABC compared with MAC, I require some revision for the publication as follows:

Major points:

1 The most important point of the article is lacking clinical relevance. What do authors know in terms of comparison between two species? It may be useful to use scoring system, result of pulmonary function test, and disease course requiring treatment etc.

2 Inclusion and exclusion criteria are not accurate: how authors defined "severe lung diseases" because some patients still have ILD, COPD, and other lung diseases in Table 1? Also, how many numbers of isolation of NTM in the four institutions? Authors should show a consort diagram for the study including numbers of inclusion and exclusion.

3 Did author include which point of CT findings for the analysis? Newly-onset or just diagnosed at each institute? Some MABC patients have a history of MAC infection or mixed infection for both species. It is important that the authors reported what stage. If patients were not treatment-naive, disease and treatment duration are helpful for readers.

4 It is helpful to show all the representative images of each radiological findings.

Minor points:

1- What proportion of M. avium and intracellulare did you include?

2- MABC showed unclassified type in 30.6%. Did authors think what this means?

3- Is there any information about A-type vs M-type vs B-type only in isolated colonies?

6. PLOS authors have the option to publish the peer review history of their article (what does this mean?). If published, this will include your full peer review and any attached files.

Reviewer #1: No

---

## [Author Response · Author response to Decision Letter 0]

1 Jun 2022

Editor Comments

The investigator’s report differences in radiographic findings in hospitalized patients found to have sputum cultures positive for either MABC or MAC.

The investigators need to address the concerns of the reviewer and provide additional information regarding the patients in the study as well as context for their findings that relate to biologic plausibility. The following need to be addressed:

1) Do the increased areas of low attenuation found in patients with MABC lung infection suggest that patients with COPD/emphysema may be more susceptible to MABC lung infection compared to MAC? Please comment in the manuscript. The results of pulmonary function tests would be helpful in this regard – it would seem that this information should be available from a retrospective chart review from at least some of the patients and needs to be provided. If COPD/emphysema is more common, what would be the basis for this?

Response: Thank you for this insightful comment. To address your concerns, we have performed an analysis on patients who were never-smokers (MABC-PD group: 20 patients, MAC-PD group: 47 patients). The findings revealed that low attenuation remained more common in patients with Mycobacterium abscessus complex pulmonary disease (MABC-PD) than in those with Mycobacterium avium complex PD (MAC-PD; 65% vs 8.5%, p<0.0001). We have specified this in the Results section (page 11, lines 149–153). This finding indicates that MABC itself (and not smoking) can cause low attenuation in the lungs. Previous reports have shown that centrilobular nodules caused by MAC infection narrow the lumens of the bronchioles at various levels [1,2]. This also leads to low attenuation due to the air trapping mechanism [3]. We presume that low attenuation was observed more commonly in patients with MABC-PD than in those with MAC-PD, because compared to MAC, MABC is highly virulent and causes more extended lesions in the lung. We have mentioned this in the revised manuscript as well (page 15 [lines 184–186] and page 16 [lines 193–203]).

We agree that pulmonary function test results would be very useful to support our conclusions. However, because this study was retrospective and cross-sectional in nature, pulmonary function tests results may not have been available for all patients. Accordingly, we did not present the results of the pulmonary function tests.

2) Why do the investigators postulate that more lung lobes are involved in patients found to have MABC lung infection? Furthermore, why do they postulate that cardiovascular disease was more common in patients with MABC cultured from their sputum?

Response: Thank you for this important comment. Previous reports have stated that MABC could be rapidly progressive and lead to lung destruction in a short period of time [4]. We presumed that the virulence intensity of MABC is related to the extensive distribution of lung lesions in MABC-PD. Thus, we mentioned that more lung lobes are involved in patients with MABC-PD. We have discussed this in the manuscript (page 18, lines 235–238).

Moreover, the reason why we stated that cardiovascular diseases were more common in MABC-PD, was because Hsu et al. pointed out that patients with either cardiovascular diseases or risk factors for cardiovascular diseases (male sex and age ≥55 years) were 4.5 times more likely to develop MABC-PD [5]. Moreover, the chronic inflammation associated with non-tuberculous mycobacteria pulmonary disease (NTM-PD) may induce a secondary cardiovascular disease. We have specified this in the manuscript (page 15, lines 177–183).

3) What are differences comparing MABC to MAC in terms of known characteristics/virulence determinants that could account for reported radiographic observations?

Response: As we have mentioned in our response above, MABC is reported to be highly virulent as compared with MAC [4]. We think that this difference may be responsible for the differences in the radiological findings between MABC-PD and the MAC-PD.

4) How many patients underwent antibiotic treatment as a result of the concern for pulmonary NTM infection? This would be an indication of the treating physician’s concern that the patient had active, progressive NTM lung infection. Although IDSA and ATS provide guidelines, the judgment of the clinician seeing the patient has relevance to the significance of the radiographic findings reported in the study. In addition, at least some follow up information with time points after initiation of treatment should be provided on all patients. It is emphasized that the treatment status of these patients is important to judge the significance of the radiographic findings.

Response: We apologize for our insufficient explanation. We actually excluded patients who received antibiotic treatment; therefore, all patients included in our study were untreated. We have revised our manuscript to clearly describe the exclusion criteria (pages 5-6, lines 82–88). Additionally, we did not follow up the clinical courses of the patients due to the study’s retrospective and cross-sectional nature. 

References for the editor

1. Fujita J. Radiological findings of non-tuberculous mycobacteria respiratory infection. Kekkaku. 2003;78:557-61.

2. Fujita J, et al. Pathological and radiological changes in resected lung specimens in Mycobacterium avium intracellulare complex disease. Eur Respir J. 1999;13:535-40.

3. Kubo K, et al. Pulmonary infection with Mycobecterium avium-intracellulare leads to air trapping distal to the small airways. Am J Respir Crit Care Med. 1998;158:979-84.

4. Victoria L, et al. Mycobecterium abscessus complex: a review of recent developments in an emerging pathogen. Front Cell Infect Microbiol. 2021;11:659997.

5. Hsu JY, et al. Mycobacterium abscessus and Mycobacterium massiliense exhibit distinct host and organ specificity: a cross-sectional study. Int J Infect Dis. 2022;116:21-6.

 

Reviewers' comments

Reviewer #1

The authors present the interesting findings about the differences between MAC and MABC pulmonary disease, focusing on CT findings. Although it is important to know the characteristics of MABC compared with MAC, I require some revision for the publication as follows:

Major points:

1 The most important point of the article is lacking clinical relevance. What do authors know in terms of comparison between two species? It may be useful to use scoring system, result of pulmonary function test, and disease course requiring treatment etc.

Response: Thank you for this comment. We completely agree with you that we should emphasize the clinical relevance of our study. During the diagnosis of non-tuberculous mycobacterial pulmonary diseases (NTM-PD), it often takes weeks or months to identify the causative NTM from respiratory specimens in clinical settings. Thus, we believe that understanding a patient’s background and radiological features suggestive of the causative NTM species will prove important for diagnosis. We have addressed this in the revised manuscript (page 4, lines 60–64). Unfortunately, we did not have the pulmonary function test results and clinical course data for all patients due to the study’s retrospective and cross-sectional nature. 

2 Inclusion and exclusion criteria are not accurate: how authors defined "severe lung diseases" because some patients still have ILD, COPD, and other lung diseases in Table 1? Also, how many numbers of isolation of NTM in the four institutions? Authors should show a consort diagram for the study including numbers of inclusion and exclusion. 

Response: We apologize for the unclear descriptions in the previous version of the manuscript. To address your concerns, we have revised the manuscript to clearly state the inclusion and exclusion criteria (pages 5–6, lines 74–93). This study included 101 patients (36 with Mycobacterium abscessus complex pulmonary disease [MABC-PD] and 65 with Mycobacterium avium complex PD [MAC-PD]).

3 Did author include which point of CT findings for the analysis? Newly-onset or just diagnosed at each institute? Some MABC patients have a history of MAC infection or mixed infection for both species. It is important that the authors reported what stage. If patients were not treatment-naive, disease and treatment duration are helpful for readers.

Response: We apologize for the unclear descriptions in the previous version of the manuscript. Accordingly, we have revised the text to clearly state the high-resolution computed tomography (HRCT) examination time points in this study. We evaluated the chest HRCT findings obtained on the day closest to the date of NTM diagnosis both 1 year before and after NTM-PD diagnosis (page 6, lines 96–98). Because we excluded patients receiving antibiotics at the time of diagnosis, all included patients were treatment-naïve. We also excluded patients co-infected with other NTM and bacteria (pages 5-6, lines 82–88).

4 It is helpful to show all the representative images of each radiological findings.

Response: To address this comment, we have included Figure 1, which is a representative image of the typical radiological findings of low attenuation in MABC-PD.

Minor points:

1- What proportion of M. avium and intracellulare did you include?

Response: Thank you for this comment. In this study, MAC consisted of M. avium (n=16, 24.6%) and M. intracellulare (n=49, 75.4%). We have specified this in the revised manuscript (page 8, lines 127–129).

2- MABC showed unclassified type in 30.6%. Did authors think what this means?

Response: We believe that this result may be explained by the virulence of MABC. Because MABC could be rapidly progressive and lead to lung destruction in a short period of time [1], it could give rise to a variety of extended lesions in the lungs; this would lead to an unclassifiable pattern on HRCT in patients with MABC-PD.

3- Is there any information about A-type vs M-type vs B-type only in isolated colonies?

Response: Unfortunately, we do not have any information on the flagged topic. Most of the MABC colonies were not preserved and could not be analyzed for subtype analyses.

Reference for the reviewer

1. Victoria L, et al. Mycobecterium abscessus complex: a review of recent developments in an emerging pathogen. Front Cell Infect Microbiol. 2021;11:659997.

---

## [Editor Report · Decision Letter 1]

6 Jul 2022

The radiological findings of nontuberculous mycobacterial pulmonary diseases: comparison between Mycobacterium avium complex and Mycobacterium abscessus complex.

PONE-D-21-40311R1

Dear Dr. Nagano,

We’re pleased to inform you that your manuscript has been judged scientifically suitable for publication and will be formally accepted for publication once it meets all outstanding technical requirements.

Kind regards,

Thomas Byrd

Academic Editor

PLOS ONE
---

## [Editor Report · Acceptance letter]

12 Jul 2022

PONE-D-21-40311R1 

Radiological findings in nontuberculous mycobacterial pulmonary diseases: A comparison between the *Mycobacterium avium* complex and the *Mycobacterium abscessus* complex 

Dear Dr. Nagano:

I'm pleased to inform you that your manuscript has been deemed suitable for publication in PLOS ONE. Congratulations! Your manuscript is now with our production department. 

Kind regards, 

on behalf of

Dr. Thomas Byrd 

Academic Editor

PLOS ONE